

# Application of multimodality perception scene construction based on Internet of Things (IoT) technology in art teaching

Haiwen Wang[1,2], Yahui Wang[1] and Juan Jin[3]

[1] School of Humanities and Arts, Macau University of Science and Technology, Macau, China
[2] School of Art and Design, Wuhan Technology And Business University, Wuhan, Hubei, China
[3] School of Economics and Business Foreign Languages, Wuhan Technology And Business University, Wuhan, Hubei, China

## ABSTRACT

Numerous impediments beset contemporary art education, notably the unidimensional delivery of content and the absence of real-time interaction during instructional sessions. This study endeavors to surmount these challenges by devising a multimodal perception system entrenched in Internet of Things (IoT) technology. This system captures students' visual imagery, vocalizations, spatial orientation, movements, ambient luminosity, and contextual data by harnessing an array of interaction modalities encompassing visual, auditory, tactile, and olfactory sensors. The synthesis of this manifold information about learning scenarios entails strategically placing sensors within physical environments to facilitate intuitive and seamless interactions. Utilizing digital art flower cultivation as a quintessential illustration, this investigation formulates tasks imbued with multisensory channel interactions, pushing the boundaries of technological advancement. It pioneers advancements in critical domains such as visual feature extraction by utilizing DenseNet networks and voice feature extraction leveraging SoundNet convolutional neural networks. This innovative paradigm establishes a novel art pedagogical framework, accentuating the importance of visual stimuli while enlisting other senses as complementary contributors. Subsequent evaluation of the usability of the multimodal perceptual interaction system reveals a remarkable task recognition accuracy of 96.15% through the amalgamation of Mel-frequency cepstral coefficients (MFCC) speech features with a long-short-term memory (LSTM) classifier model, accompanied by an average response time of merely 6.453 seconds—significantly outperforming comparable models. The system notably enhances experiential fidelity, realism, interactivity, and content depth, ameliorating the limitations inherent in solitary sensory interactions. This augmentation markedly elevates the caliber of art pedagogy and augments learning efficacy, thereby effectuating an optimization of art education.

Corresponding author
Juan Jin, 20081008003@wtbu.edu.cn

# INTRODUCTION

With the arrival of the 5G network era, the deep integration of emerging technologies such as the Internet of Things (IoT), big data and artificial intelligence with education and teaching has promoted the practical process of intelligent education research. New changes have occurred in teaching equipment, teaching tools, teaching methods, *etc*. Art teaching has also made breakthroughs. The IoT is the application integration of sensor technology, communication technology and information technology in the 21st century. It is the third wave of the world's information industry after computers, the Internet and mobile communication networks (*Cui et al., 2021*). The concept of the IoT can be traced back to 1999. The Massachusetts Institute of Technology's automatic identification laboratory uses various information sensing devices such as network radio frequency identification (RFID), infrared sensors, global positioning devices, *etc*. According to the protocol, any object can access the Internet for information exchange and communication. The network can complete tasks such as intelligent identification, positioning, tracking, and monitoring of targets (*Liu et al., 2021*). In recent years, IoT technology has been widely used in education. *Rani, Dhrisya & Ahalyadas (2017)* developed an AR-based interactive gesture application on mobile devices and simulated physical experiments in virtual classrooms. *Dave, Chaudhary & Upla (2019)* combines the IoT technology with gestures, uses adaptive hand segmentation and pose estimation methods to recognize defined gestures, and simulates the experiment process with real hands in virtual experiments. *Karambakhsh et al. (2018)* and *Liu et al. (2019)* used gesture, voice, holographic interaction and viewpoint-tracking technology to apply in actual teaching, thoroughly verifying the vast potential of gesture application in medical learning and education systems. *Cao & Liu (2019)* proposed an interactive teaching system in which students naturally and directly manipulate 3D objects through gesture interaction and intuitively explore the spatial relationship between spheres and polyhedrons, which has the advantages of easy use and attractiveness. *Mensah, Odro & Williams (2023)* developed the AR geometry teaching system with gesture operation, implemented and evaluated the system, and verified the individual differences in students' 3D thinking skills and the importance of creating a personalized, dynamic, intelligent learning environment. *Westerfield, Mitrovic & Billinghurst (2015)* combines the sensor equipment with the education process, and research has found that users' interaction efficiency has improved.

Art education can lead students to establish correct aesthetic values, cultivate noble moral sentiments, stimulate imagination and innovative consciousness, and promote students' healthy growth and all-round development. However, in the current teaching of art courses in schools, there are generally the following problems: First, the presentation of teaching content is relatively dull and simple. Teachers often present teaching content through words and pictures, lacking dynamic and immersive explanations, which easily leads to visual fatigue and distraction of students. Second, the teaching process lacks real-time interaction, and students often receive knowledge and skills passively, and their thinking and imagination cannot reach enlightenment. The lack of timely feedback in the teaching process will also affect students' enthusiasm and ignore individual differences. In

recent years, the integration of IoT technology and art education has made art teaching methods more innovative and teaching content more colorful, effectively solving the above problems.

Leveraging IoT technology to construct a multimodal perceptual environment for conducting art classroom instruction, communication, and presentation accentuates the distinctive facets of art education. This initiative aims to enhance students' creativity and capacity for aesthetic critique, effectively fostering their creative ideation and imagination while advancing the development of the art education discipline. *Nathoo et al. (2020)* underscored the efficacy of IoT technology in augmenting students' learning engagement and practical competencies within art education. Similarly, the collaborative learning framework elucidated by *Welagedara, Harischandra & Jayawardene (2021)* enabled students and educators to engage in role-playing and collaborative musical rehearsals within a virtual setting, markedly enhancing students' expressive capabilities. *Tran (2023)* innovatively devised a stage music interactive experience apparatus, wherein the pianist's performance triggers dynamic visual alterations in the surrounding line lights, engendering a more vibrant aesthetic rendition. Additionally, tactile interaction with real plants integrated into the apparatus allows visitors to evoke distinct sounds and hues in a virtual realm, thereby elucidating the botanical growth process in a virtual and abstract manifestation (*Niu & Feng, 2022*).

The integration of IoT technology and art education makes art teaching methods more innovative, teaching content more colorful, and makes teachers and students, students and students closer. The multimodality integration of vision, hearing, and touch will enrich students' aesthetic experience.

Multimodal recognition encompasses recognizing expressions, gestures, vocalizations, posture, sensorial inputs, and additional modalities. In expression recognition, *Grafsgaard et al. (2013)* delineated that facial dynamics are prognosticators of students' engagement, mood states, and learning efficacy through the meticulous tracking of nuanced facial gestures. Notably, subtle movements such as eyebrow furrowing, eyelid tension, and mouth corner downturns have been identified as significant indicators. Of particular relevance, the degree of mouth-corner depression emerges as a potent predictor of learning outcomes and self-reported performance. *Whitehill et al. (2014)* explored a method to identify learning participation from students' facial expressions automatically. This method uses SVM with Gabor features to detect students' learning participation every 10 s in the video stream. *Yang et al. (2018)* proposed a remote classroom learning state detection method based on facial expression recognition. He trained the models of six basic expressions (happy, surprised, afraid, sad, angry and disgusted) defined on the JAFFE dataset. He experimented with them as the basic learning state of students. In the research of speech recognition, most researchers use machine learning algorithms such as Gaussian mixture model (*Patel & Kushwaha, 2020*), support vector machine (*Jun, 2021*), hidden Markov model (*Mor, Garhwal & Kumar, 2021*), naive Bayes (*Wang et al., 2017*), k-nearest neighbor (*Pao et al., 2007*) for speech emotion recognition, and have done a lot of research work. In the research of gesture recognition, gesture recognition methods are mainly divided into neural network-based gesture classification and recognition methods (*Su et*

*al., 2021*), hidden Markov model (HMM) based gesture recognition (*Sinha et al., 2019*) andgeometric feature-based gesture recognition (*Pisharady & Saerbeck, 2015*). However, the current teaching mode based on IoT technology mostly stays on vision and voice, lacking direct stimulation of users' limbs and senses. Therefore, more interaction channels need to be explored to give different information forms to users participating in the interaction. This study focuses on organically combining the architecture of multisensory channel interaction with art teaching. The main contributions are as follows:

Development of a multimodal perception system: The research introduces a novel approach to address the deficiencies in art education by creating a multimodal perception system using IoT technology. This system incorporates sensors to capture students' images, voice, location, motion, ambient light, and other data, facilitating real-time interaction during teaching sessions.

Advancement in technology for art education: The study advances critical technologies such as visual feature extraction utilizing the DenseNet network and voice feature extraction *via* the SoundNet convolutional neural network. These advancements enable a more comprehensive and immersive learning experience by integrating multiple sensory channels, thereby establishing a new art teaching model that emphasizes visual information while engaging other senses as supporting participants.

Enhancement of learning efficiency and experience: The research evaluates the usability of the multimodal perceptual interaction system and demonstrates its effectiveness with a task recognition accuracy of 96.15% and an average response time of only 6.453 s. This enhancement significantly improves art instruction's experience, realism, interactivity, and content richness, ultimately optimizing art education and learning efficiency.

## RELATED WORK

Modality is the interaction mode between humans and the external environment established through sensory organs, such as vision, hearing, touch, smell, taste, *etc*. Multimodality human–computer interaction refers to the information interaction between computers and people's multiple sensory modes. *Stein (2012)* put forward the "multimodality teaching method", called the multimodality teaching method, by changing the traditional single-mode teaching mode into a method of communication using multimodality methods. The characteristic of the multimodality teaching method is that classroom communication goes beyond language. Language and characters are only two of the ways of communication. Other modes, including voice, color, image, action and gesture, can convey meaning comprehensively (*Lv et al., 2022*). *Stein (2012)* believes that the characteristic of the multimodality teaching method is that classroom communication goes beyond language. Language and words are only two of the ways of communication. Other modes, including voice, color, image, action and gesture, can convey meaning comprehensively (*Alzubi et al., 2023*).

In recent years, the construction of multimodality perceptual scenarios based on IoT technology has gradually become a hot topic in intelligent education. *Romero, Bobkina & Radoulska (2018)* asked students to use the modes of language, vision, audio,

gesture and space to convert text-based materials into multimodal writing courses and found that students' understanding of multiculturalism was deepened due to the integration of other abilities (such as painting, computer programming, video editing and storytelling) and language. *Sharma & Giannakos (2020)* believed that the difference between multimodality and traditional teaching methods lies in emphasizing the interaction between multiple modes. At the same time, they emphasized that teachers must select modes and combinations in a targeted way to give full play to the greatest advantage of the multimodality teaching method (*Sharma & Giannakos, 2020*). *Yunita et al. (2022)* proposed that the multimodality teaching method is based on communication, including language, gesture, body, space, image, text, *etc*. Context awareness uses intelligent sensing devices to intelligently perceive humans, machines, objects and other physical elements under specific space–time conditions, obtain useful feedback information for users, and achieve interactive integration between users and the environment with computing devices through data analysis and processing. *Tortorella & Graf (2017)* use sensors to collect students' image data, voice data, location data, motion data, ambient light data, etc., to model learning situation information and recommend appropriate multimedia learning resources for learners in the adaptive mobile learning system according to learners' learning style and learning situation information. Constructing multimodality perception scenarios through IoT technology is conducive to stimulating students' sense channels, such as vision, hearing, and touch, and stimulating their learning interest and creative desire. At the same time, it makes the presentation of the teaching content more visual, increases the interest of learning, attracts students' attention, and effectively conveys knowledge. Through IoT technology, we carry out multimodality teaching, interact with students through graphics, light, sound, and color, expand students' thinking, and cultivate their exploration ability. The process involves real-time interaction, creating a relaxed learning environment for students and increasing the heat and interaction of learning.

The multimodality teaching method is based on social semiotics, multimodality discourse analysis and other theories. It combines the students' vision, hearing, touch and other senses by using external video, sound and other symbols to cooperate better to complete the communication and transmission of teaching content. The complexity of the education situation brings great challenges to the development of intelligent education-related research. How to model the education situation accurately and realize its computability is the key problem that relevant researchers in the field of intelligent education need to solve in the future. A multimodality perception system based on IoT technology is built to meet the needs of art teaching. Through multiple interaction channels such as vision, hearing, touch, and smell, temperature, humidity, vibration, body sense and other sensing means are integrated to collect students' image, voice, location, motion, ambient light and other data, model the learning situation information, and arrange the sensors in a reasonable space to make the interaction process simple and natural. Then, take the digital art flower cultivation interaction as an example, design a multisensory channel interaction task, break through the key technologies such as visual feature extraction based on DenseNet network and voice feature extraction based on SoundNet convolution neural network, and realize a new art teaching mode that takes visual information as the

dominant factor and other senses as the auxiliary participants. Finally, this study evaluated the usability of the multimodality perceptual interaction system, and the results showed that the system had significantly improved in the aspects of experience, realism, interactivity, and content richness, which made up for the unnatural single sensory interaction, improved the teaching quality and learning efficiency of art teaching, and achieved the goal of optimizing art teaching.

# DESIGN AND IMPLEMENTATION OF A MULTI-MODE PERCEPTION SYSTEM BASED ON IOT TECHNOLOGY

## Image feature extraction

Facial expression recognition can analyze students' learning dynamics and teachers' teaching status, judge classroom emotions, and objectively evaluate classroom effects. Facial expression recognition technology for classroom interaction and emotional evaluation has become a hot issue in intelligent education, which has significant exploration and application value. This study constructs a multimodal perceptual scene, using facial expression recognition as one of the interaction methods, which can recognize the expressions of teachers and students and analyze and evaluate the classroom teaching situation according to the recognition data to adjust the teaching strategy.

This research uses the DenseNet-BC convolution neural network to extract image features. DenseNet network adopts a dense connection mode. It does not need to re-learn redundant feature mapping. It has the advantages of reducing gradient disappearance, strengthening feature transmission, and using features efficiently. DenseNet-BC network includes the Bottleneck layer and the Transition layer. The transition layer is a network layer composed of roll-up and pooling layers. DenseNet-BC network includes three Dense Blocks and two transition layers. Figure 1 shows the structure of the DenseNet-BC network, where Dense block n represents the nth dense block.

This study selected a variety of classifiers for experiments, including classical classifiers such as support vector machine (SVM) and random forest (RF). At the same time, considering that when video samples are divided into image samples, extended sequence image frames are obtained, and long-short-term memory (LSTM) has significant advantages in processing long sequence data, a classifier based on LSTM is designed for the experiment. The structure of the classifier based on LSTM is shown in Fig. 2. The input sequence X is a feature of different times. A batch standardization layer is added after the input layer. The LSTM array composed of multiple LSTM nodes captures the feature information. The feature information of different times is averaged through the average pooling layer and output to the Softmax layer for classification.

In multimodality emotion recognition, weighted voting methods and weighted average methods are common decision fusion methods. Among them, the voting method is more suitable for the case where the models in decision fusion are independent. Considering that the training of each model in this study is independent and there is no strong dependence, using the weighted voting method for decision fusion may bring some improvement. The weighted voting method is implemented explicitly as follows:

Conv 1     Dense block 1     Conv 2   Pool 1     Dense block 2     Conv 3   Pool 2     Dense block 3     Pool 3

**Figure 1** **DenseNet-BC network structure.**

Suppose the number of expression categories is M, the number of models is L, $h_i$ is the $i^{th}$ emotion recognition model, and $w_i$ is the contribution weight of the ith model to the decision results of the fusion model, where the constraints of $w_i$ are:

$$\begin{cases} \sum_{i=1}^{L} w_i = 1 \\ w_i \in [0,1], i = 1,2,\ldots,L \end{cases} . \tag{1}$$

For sample x, let f(x) be the set of weighted voting values of various expression categories based on the weighted voting method, and y(x) be the decision result of expression categories, then:

$$f_j(x) = \sum_{i}^{L} w_i I\left(h_i(x),j\right) . \tag{2}$$

$$y(x) = argmax\, f_j(x)$$

where $j = 1,2,\ldots,M$, the definition of indicating function I is:

$$I(a,b) = \begin{cases} 1, a=b \\ 0, a \neq b \end{cases} . \tag{3}$$

This research is mainly based on the AFEW data set for multimodality emotion recognition research. Firstly, the video data of the AFEW dataset is divided into audio and picture data, and emotional features of voice and image modes are extracted, respectively. When extracting image features, we must first perform two preprocessing operations: face detection, correction, and clipping. Since the author of the AFEW dataset has provided most of the cropped grayscale facial images, the image data not provided are only 17 videos in the Train training set and 12 videos in the Val verification set. Therefore, we only preprocess the image data that is not provided. After successfully extracting the face gray image, we still need to carry out histogram equalization to reduce the impact of light on the image. After the preprocessing operation is completed, we fine-tune the pre-trained DenseNet-BC convolutional neural network model on the FRE2013 dataset and then extract image features using the preprocessed image as the input of the fine-tuned model. When using the DenseNet-BC convolution neural network to extract features, take the output of the last average pooling layer of the DenseNet-BC network as the feature, which is marked as DenseNet-pooling3.

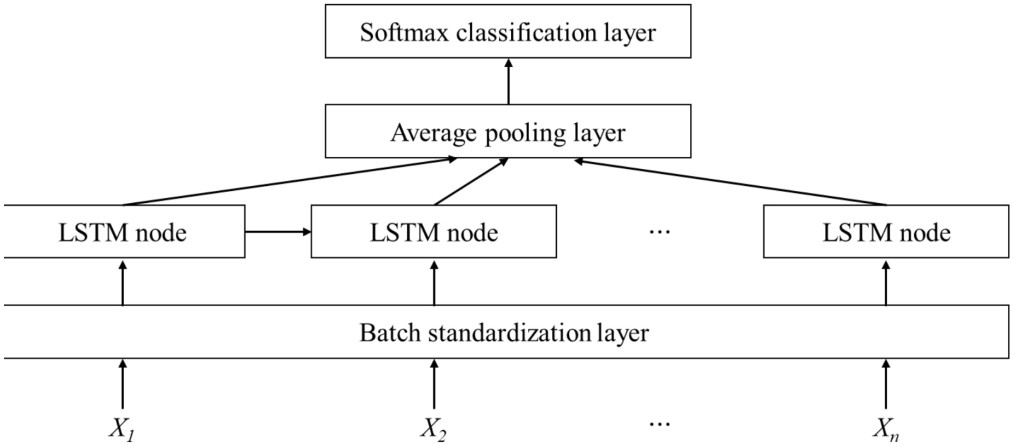

**Figure 2  Structure of LSTM based classifier.** This image was made by the authors without any data source.

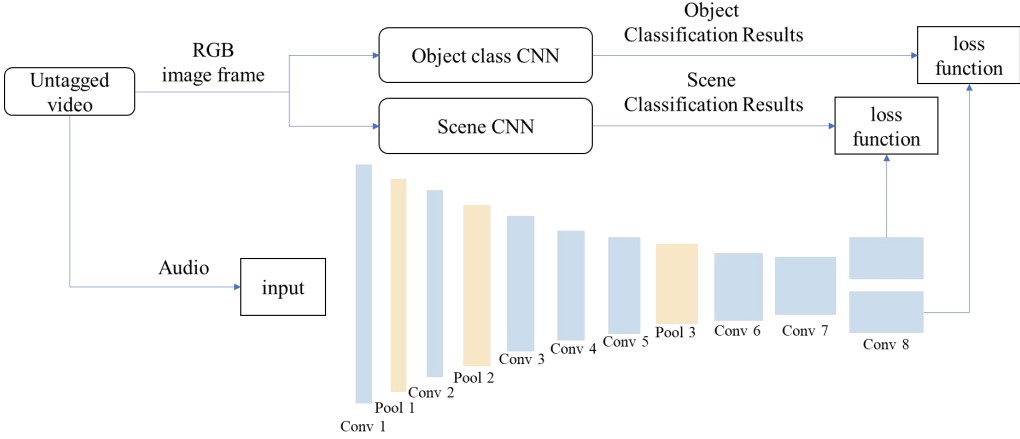

**Figure 3  SoundNet network structure and implementation schematic diagram.**

## Speech feature extraction

SoundNet is a deep convolutional neural network with a high learning ability of voice information (*Aytar, Vondrick & Torralba, 2016*). The structure and implementation schematic diagram of the SoundNet network are shown in Fig. 3.

First, video is cut into audio and RGB image frames. RGB image frames are recognized and classified by using the image-like convolutional neural network (ImageNetCNN) and scene-like neural network (Places CNN), respectively, and the results of RGB image frame classification are used as the supervision information of SoundNet network, from which the voice-related information can be learned. The SoundNet network used in this study consists of eight layers of convolution layer and three layers of pooling layer, and the loss function is KL divergence.

Before extracting speech features, the SoundNet network preprocesses audio files through three essential operations: resampling, framing, and windowing. Resampling standardizes the sampling rate across all audio inputs, ensuring uniformity for subsequent processing steps. Framing divides the audio signal into short, overlapping segments to capture temporal dynamics effectively, preserving transient sounds and temporal information. Finally, windowing applies a window function to each frame, such as the Hamming window, to minimize spectral leakage and enhance the accuracy of spectral analysis by smoothing frame edges and reducing artifacts. Together, these preprocessing steps ensure the audio data is appropriately structured and represented for extracting meaningful speech features, enhancing the network's ability to understand and process audio inputs effectively.

When using the SoundNet convolution neural network to extract features, the original data of the audio file is taken as the input, and the extracted features are marked as SoundNet. The SoundNet network structure and implementation schematic diagram used in this study are shown in the figure, where Conv n represents the nth layer of convolution, and pool n represents the nth layer of pooling.

Configuring the Mel filter bank involves determining the parameters of each filter in the bank based on the Mel scale. The Mel filter bank aims to mimic the human auditory system's response to different frequencies. Each filter in the bank is defined by its center frequency, which is spaced logarithmically along the Mel scale. The extraction process of Mel frequency cepstrum coefficient MFCC is as follows: First, FFT is performed on a sampled frame of discrete speech sequence $x(n)$. The formula of FFT is as follows:

$$X(k) = \sum_{n=0}^{N-1} x(n)e^{-j\frac{2\pi}{N}nk}, k = 0, 1, 2, \ldots, N-1 \tag{4}$$

where N is the frame length.

Configure the Mel filter bank and calculate the filtered output. The frequency response Hm(k) of the Mel filter is:

$$H_m(k) = \begin{cases} 0, k < f(m-1) \\ \dfrac{2(k-f(m-1))}{(f(m+1)-f(m-1))*(f(m)-f(m-1))}, f(m-1) \leq k \leq f(,) \\ \dfrac{2(f(m+1)-k)}{(f(m+1)-f(m-1))*(f(m)-f(m-1))}, f(m) \leq k \leq f(m+1) \\ 0, k \geq f(m+1) \end{cases} \tag{5}$$

where f(m) is the center frequency of the filter.

Then, calculate the logarithmic energy S(m) output by each filter bank.

$$S(m) = ln\left(\sum_{k=0}^{N-1} |X_a(k)|^2 H_m(k)\right), 0 \leq m \leq M \tag{6}$$

where, M is the number of filters. $X_a(k)$ represents the input signal in the frequency domain.

Finally, MFCC coefficient C(n) can be obtained by DCT of discrete cosine transform, and the formula is described as follows:

$$C(n) = \sum_{m=0}^{N-1} S(m) \cos\left(\frac{\pi n(m-0.5)}{M}\right), n = 1, 2, \ldots, L \tag{7}$$

where $L$ is the MFCC coefficient's order, *cos* denotes the cosine function. This process typically results in a set of MFCC coefficients representing the features of the input signal that are relevant for speech or audio processing tasks.

## Sensor technology

This research uses temperature, humidity, somatosensory control, vibration, and other types of sensors to build a multimodal perception scene. Each sensor corresponds to different senses of people to obtain various information and provide interaction interfaces between users and systems.

Through digital temperature and humidity sensors, analog signals are measured and directly converted into digital signals, which are used to measure and control air temperature and humidity information in the multi-mode perception scenario. The Kinect body sensor is used to capture gesture information from users. Kinect uses a camera to scan the real environment as a whole. In the digital art flower cultivation system, the body sensor captures the user's flower-picking action. Then, the system simulates the scene where the user touches the flower and makes the petals fall. The odor generator can feed the corresponding odor to the user according to different scenes, stimulate the user's sense of smell, and deepen their perception of the screen. In the digital flower culture system, the somatosensory control sensor recognizes the user's hand movements and then triggers the aroma generator to work, releasing the fragrance of flowers. Vibration and sound feedback sensors provide users with tactile and auditory stimuli, including playing prompt sound effects and outputting vibration. In the digital flower cultivation system, the sound feedback sensor outputs the preset process guidance sound effect during the user's participation in the course tasks, such as "Welcome to the art teaching class. Do you want to cultivate a rose?", "Don't worry, water the soil first; the seeds will sprout!", and so on. It can also output natural sound effects. For example, when the flowers are in full bloom, they can output the sound effects of bees flying to increase the information feedback on the user's hearing so that users can immerse themselves in it. The pyroelectric infrared sensor is used to detect the existence of human beings by detecting infrared rays. In the digital art flower cultivation system, the system will automatically identify the user's position and give feedback on things such as breeze, fragrance, and vibration, according to the scene settings. Each sensor module is shown in Fig. 4.

## Construction of multimodality perceptual scene system

The realization of multisensory interaction is to achieve reasonable participation in interactive behavior through reasonable design of interactive tasks and finally form a good interactive experience. The design of interactive tasks is the first problem to be solved in the multisensor interaction model. Interactive task design needs to arrange task flow and

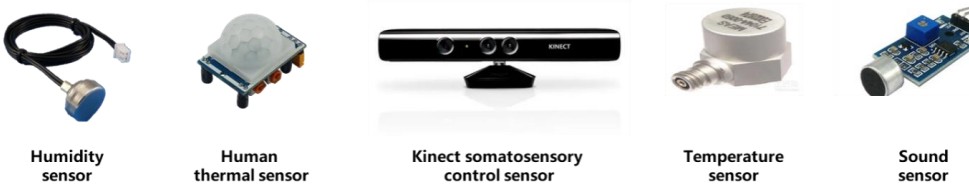

| Humidity sensor | Human thermal sensor | Kinect somatosensory control sensor | Temperature sensor | Sound sensor |

**Figure 4 Multi type sensors to build multi-mode perception scenarios.** This images was made by the authors without any data source.

specific ways of task participation according to the impact on integrating participants' sensory channels. The system controls the visual channel so that the channel integration process presents the effect of visual information leading and other sensory assistance participating. In addition, the system needs to design the auditory channel information, including system prompt tone, music, sound, *etc.*, and the time and location of the sound generation device.

This research builds a multimodality perception situational teaching system based on IoT technology. Students can immerse themselves in the art classroom through visual, auditory, tactile, olfactory and other interactive channels. Following the organization mode of human–computer interaction, the sensors are arranged in a reasonable space. The sensors correspond to all human senses, making the interaction process simple and natural. This research takes the interactive prototype system of cultivating digital art flowers as an example. The digital art flower cultivation system includes water level sensors for watering amounts, temperature and humidity sensors for acquiring temperature and humidity signals, and wind sensors for acquiring wind direction signals. The temperature sensor receives temperature data and controls seed germination; the humidity sensor obtains the humidity data and controls the bud to blossom; the wind sensor receives wind data and controls the movement direction of digital flowers.

The display device outputting visual information should be placed in the front, the odor-generating device outputting odor should be placed in the front of the user and lowered, the audio device outputting sound on both sides of the display model screen should be set and the somatosensory control device outputting gesture and other operations should be placed in the front and close to the user; each sensor of input data should be distributed at the place where people can reach their left and right hands. This spatial layout allows users to participate in multisensory interaction naturally and comfortably and obtain a more natural interaction experience than single-sensory interaction.

According to the relationship between the task and the system, a system state transition diagram can be obtained, as shown in Fig. 5. The yellow diamond represents the decision maker of information fusion. In the input phase, the decision maker completes the accurate judgment of the input sensory information, triggers the system to switch states, and makes decisions on the output content and output time weight of each state according to the input information to accurately transmit the information to the user's senses, The condition of

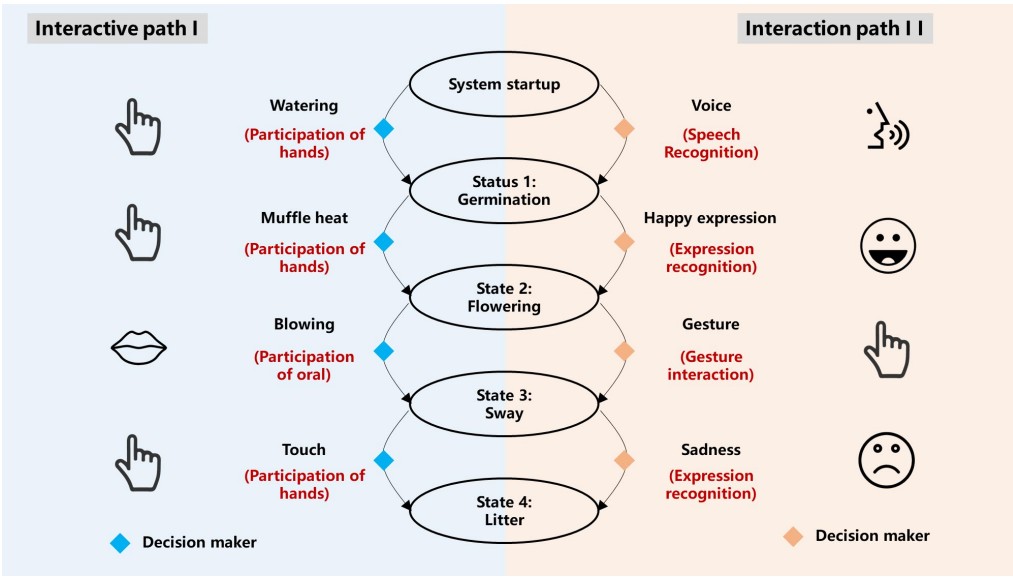

**Figure 5** **System state transition diagram.** This image was made by the authors without any data source.

**Table 1** Interactive task setting for cultivating digital art flowers.

|  | Task number | Mission purpose | Task action | Task object | Task results |
|---|---|---|---|---|---|
| Interactive path I | 1 | seed germination | Watering | Water level sensor | See the seeds sprout and hear the sound of watering |
|  | 2 | Bud blossoming | Muffle heat | Temperature sensor | See the flowers, smell the flowers, hear the bees coming |
|  | 3 | Flower swaying | Blowing | Wind direction sensor | See the flowers swing |
|  | 4 | Petals falling | Touch | Body sensor | See the petals fall |
| Interaction path 2 | 1 | seed germination | Voice | Speech recognition | See the seeds sprout and hear the sound of watering |
|  | 2 | Bud blossoming | Happy expression | Expression recognition | See the flowers, smell the flowers, hear the bees coming |
|  | 3 | Flower swaying | Gesture | Gesture recognition | See the flowers swing |
|  | 4 | Petals falling | Sad expression | Expression recognition | See the petals fall |

sensory channel integration is formed, which leads to channel integration and completes the interaction process.

The interactive task of students' participation in training digital art flowers is shown in Table 1.

The interactive task of cultivating digital art flowers designed in this study includes two paths. The specific steps of path one include: through watering action, students trigger the water level sensor to make the seeds germinate; through the action of covering heat, students trigger the temperature sensor to make the buds bloom; through the blowing action, students can trigger the wind direction sensor to make the flowers swing; by touching, students can trigger the body sensor to make the petals fall. The specific steps of

**Table 2  Accuracy of facial expression recognition model based on image features.**

| Model No | Features | Classifier | Accuracy |
|---|---|---|---|
| 1 | DenseNet-pooling3 | LSTM | 95.23% |
| 2 | DenseNet-pooling3 | SVM | 90.15% |
| 3 | VGG-conv13 | LSTM | 88.23% |
| 4 | VGG-conv13 | SVM | 83.81% |
| 5 | VGG-fcl | LSTM | 86.63% |
| 6 | VGG-fcl | SVM | 81.21% |

Path 2 include sprouting seeds through voice interaction, making buds blossom through recognizing smiling expressions, making flowers swing through gesture recognition, and dropping petals through sad expressions. When the seeds germinate, the students will hear the sound of watering. When the buds bloom, the sensor signal is transmitted to drive the motor to rotate. At the same time, the spice box is opened, and the motor blows air into the spice box to diffuse the gas molecules. The smell is transmitted to the people. At this time, you can see the flowers blooming, smell the flowers, hear the bees flying, and see the petals falling one by one when the petals fall.

## RESULT ANALYSIS AND DISCUSSION

The experiments described in this article were conducted on a Windows 10 Professional operating system, utilizing an AMD Ryzen5 3500U 2.10 GHz processor and 8GB of memory. The experiments used Python 3.7 and the PyCharm 2019.3 integrated development environment.

### Evaluation of image and speech recognition models

After model training and testing, the classification results of facial expression recognition based on this study's DenseNet-BC convolution neural network are shown in Table 2.

The analysis of experimental results reveals a noteworthy improvement in accuracy, approximately 5% when comparing the LSTM-based classifier with the support vector machine (SVM) classifier utilizing image features—DenseNet pooling 3, VGG conv13, and VGG fc1. Notably, the highest accuracy in expression classification is observed with the DenseNet pooling features. Consequently, this study combines DenseNet pooling3 features and the LSTM classifier to underpin the multimodal perceptual scene construction algorithm.

Expanding on speech features involving MFCC and SoundNet, both SVM and LSTM-based classifiers are employed for speech recognition purposes. The classification outcomes are presented in Table 3.

The proposed LSTM+SVM model presents distinct advantages in art education, notably surpassing traditional classifiers with a notable 6% accuracy improvement, particularly evident when analyzing speech features like MFCC and SoundNet. By incorporating MFCC speech features with the LSTM classifier, the model achieves a remarkable accuracy rate of 96.15%, showcasing its ability to integrate multimodal information for enhanced scene construction effectively. This combination demonstrates the LSTM's prowess in capturing

**Table 3  Speech recognition model and its accuracy.**

| Model No | Features | Classifier | Accuracy |
|---|---|---|---|
| 1 | MFCC | LSTM | 96.15% |
| 2 | MFCC | SVM | 90.95% |
| 3 | SoundNet | LSTM | 89.36% |
| 4 | SoundNet | SVM | 82.84% |

temporal dependencies and nuanced patterns within data. It underscores its suitability for tasks requiring complex scene understanding, which is essential in art education contexts. Moreover, the LSTM's capacity to handle temporal dynamics and adapt to varying input sizes and resolutions further solidifies its position as a superior choice for applications necessitating nuanced analysis and comprehension of artistic expressions.

## Usability evaluation of multimodality perceptual interaction system

Usability evaluation experiments are devised to assess various indicators across each stage of users' engagement with the multimodal awareness situational system. These indicators encompass user operation time, frequency of actions, occurrence of errors, system response time, and instances of system exceptions. The objective is to evaluate the system's responsiveness, feedback mechanisms, fault tolerance, ease of use, and facilitation of learning, among other aspects.

Responsiveness and feedback can be quantified as response time and feedback data; fault tolerance can be quantified as the number of system exceptions and user errors; low difficulty can be quantified as the number of movements; easy to learn can be quantified as the time interval between system prompts and user operations to verify the availability of multimodality perceptual situational systems.

- User learning time $t_1$: Start time of user watering operation—end time of prompt tone;
- System response time $t_2$: Prompt tone start time—start time of user watering operation;
- User learning time $t_3$: Start time of user heating operation—end time of prompt tone;
- System response time $t_4$: Prompt tone start time—start time of user's hot operation;
- User learning time $t_5$: Start time of user blowing operation—end time of prompt tone;
- System response time $t_6$: Observe whether the system reacts immediately after blowing to indicate the responsiveness of the system to the task;
- Number of system exceptions: Number of system non-responses+number of system response errors.

Based on the data of 20 experimenters, this study used SPSS data analysis software to analyze the original data results, as shown in Table 4. Finally, it calculated the average learning time of users, the average response time of the system and the corresponding standard deviation, as shown in Table 5.

By analyzing the mean and standard deviation, we can determine the system's low difficulty, easy learning habits, and fault tolerance. Based on the analysis of the above data results, we can draw the following conclusions: first, in terms of the average learning time of users, the system relies on its voice prompts to inform users of the operation method.

**Table 4  Statistical table of results obtained by SPSS software.**

|  | User learning time $t_1$ | User learning time $t_2$ | User learning time $t_3$ | System response time $t_4$ | User operands |
|---|---|---|---|---|---|
| Mean value | 1.5930 | 1.2770 | 1.5630 | 6.4530 | 3.1000 |
| N | 20 | 20 | 20 | 20 | 20 |
| Standard deviation | .1561 | .1028 | .2719 | .1290 | .3077 |

**Table 5  Statistics of average user learning time and average system response time.**

| Test content | Average learning time of users | Standard deviation A | Average system response time | Standard deviation B |
|---|---|---|---|---|
| 1. Let the seed germinate | 1.593 s | 0.1561 | 6.453 s | 0.1290 |
| 2. Let the buds bloom | 1.277 s | 0.1028 | 5.451 s | 0.2023 |
| 3. Let the flowers swing | 1.563 s | 0.2720 | 3.234 s | 0.0123 |
| 4. Let the flowers wither | 1.732 s | 0.2031 | 4.231 s | 0.2123 |

**Table 6  Statistics of average user operation errors and average system exceptions.**

| Average total number of user operations | Standard deviation C | Average number of user operation errors | Average number of abnormal conditions of the system |
|---|---|---|---|
| 3.10 | 0.3078 | 0.10 | 0.09 |

The user's learning time is within 2s, and the standard deviation is minimal. The data fluctuation is very small, which indicates that a large number of users' learning time is close to the average value, and the user's learning cost is meager, so it can be seen that the system is easy to learn. In terms of the average response time of the system, the average response time of the task is 6.453s, and the standard deviation is 0.129, indicating that the system response time on this task is stable at about 6.45s, with slight fluctuation, which ensures the stability of the system response time and the continuity of user operations. The user's subjective evaluation results also reflect that the user is delighted with selecting this time threshold. By observing whether the user's blowing behavior immediately caused the system response of flower swaying, the results showed that 20 testers performed well in the task.

Table 6 shows the average user operation errors and system exceptions. The average number of users' operations is 3.1, which reflects the low difficulty of the system, while the average number of system exceptions is only 0.09, which reflects the strong fault tolerance of the system.

In this study, the subjective evaluation scale evaluates the realism, interactivity and content richness of the multimodality perception scenario system. The scoring results of each indicator are shown in Table 7.

From the statistical results of the subjective evaluation scale, the multimodal perception scenario system strengthens the expression of information, enriches the level of information, makes up for the lack of single sensory information, improves the user's sense of pleasure,

**Table 7   The subjective evaluation effect scale of realism, interactivity and content richness.**

| Effectiveness scale indicators | Average score |
|---|---|
| The multi-modal perception scene system gives me a very comfortable experience | 4.4 |
| Flowers generated by multimodality perception scene system I feel very real | 4.6 |
| The interaction of multimodality perceptual scene system is very strong | 4.6 |
| I feel natural using multimodality perceptual situational systems | 4.1 |
| The feedback from the multi-modal perception scene system is very rich | 4.7 |
| I can smell the flowers when I see them blooming | 4.8 |
| Total score | 27.2 |

and achieves a more realistic and natural user experience. Adding olfactory channel feedback enhances flowers' authenticity and the system's interactivity. This shows that the multisensor channel stimulation combined with the visual and auditory sense and olfactory sense of the system brings users a better sensory experience. The smell generation function has significantly improved the user experience of the whole system. Overall, it can be concluded that the system's information authenticity, interactivity and content richness are good.

## CONCLUSION

The experimental findings delineated in this manuscript showcase the efficacy of the devised model in alleviating the challenges associated with the monotonous dissemination of art teaching materials and the absence of real-time interaction inherent in the pedagogical process. By strategically situating sensors within conducive environments and harnessing the symbiotic exchange of information across disparate modalities, interactive endeavors are meticulously crafted to enkindle visual, auditory, tactile, and olfactory avenues of engagement. Such an approach conscientiously attends to the organic cadence of student involvement in art education, fostering an immersive ambiance within the art studio. Pioneering advancements in pivotal technologies, such as extracting visual features *via* DenseNet networks and audio characteristics through SoundNet convolutional neural networks, furnish the algorithmic backbone for crafting multimodal perceptual panoramas. This engenders a paradigm shift in art education, primarily underpinned by visual stimuli augmented by ancillary sensory modalities. The outcomes of usability assessments predicated on the multimodal perception interaction system evince marked enhancements in experiential realism, interactivity, and content richness, effectively rectifying the constraints of contrived single-sensory interactions and augmenting the pedagogical efficacy and learning proficiency in art education. Consequently, this engenders the optimization of educational objectives in art.

While the instantiation of multimodal perceptual frameworks heralds a pioneering avenue for interactive art pedagogy, it is imperative to uphold a steadfast commitment

to substantive content, artistic ethos, and cultural significance. A reasonable balance must be struck to prevent undue preoccupation with sensory gratification, eschewing an overemphasis on technological novelty, sensory indulgence, and interaction per se. This study, epitomized by the digital art flower cultivation interaction, introduces a task emblematic of multisensory channel interaction. Nonetheless, the task's inherent simplicity and lack of artistic expression underscore avenues for further refinement in scholarly pursuits. Future inquiries should delve into more intricate and artistically nuanced endeavors, integrating cultural motifs, augmenting human–computer interaction paradigms, and enriching the aesthetic repertoire. Moreover, evaluations should extend to encompass longitudinal impacts, while ethical considerations necessitate prudent deliberation to ensure a holistic approach that privileges both technological innovation and artistic expression. Such endeavors are indispensable for optimizing educational efficacy and proficiency in art.

### Funding
This work was supported by The Special Fund of Advantageous and Characteristic disciplines(Group) of Hubei Province. The funders had no role in study design, data collection and analysis, decision to publish, or preparation of the manuscript.

### Grant Disclosures
The following grant information was disclosed by the authors:
The Special Fund of Advantageous and Characteristic disciplines(Group) of Hubei Province.

### Competing Interests
The authors declare there are no competing interests.

### Author Contributions
- Haiwen Wang conceived and designed the experiments, performed the experiments, performed the computation work, prepared figures and/or tables, authored or reviewed drafts of the article, and approved the final draft.
- Yahui Wang performed the experiments, analyzed the data, performed the computation work, authored or reviewed drafts of the article, and approved the final draft.
- Juan Jin conceived and designed the experiments, analyzed the data, performed the computation work, prepared figures and/or tables, and approved the final draft.

### Data Availability
The code is available in the Supplemental File.
The data is available at Zenodo: Hong, Y. (2022). annotation data for art painting detection and identification (Version v1) [Data set]. Zenodo. https://doi.org/10.5281/zenodo.6551801.

## Supplemental Information

Supplemental information for this article can be found online at http://dx.doi.org/10.7717/peerj-cs.2047#supplemental-information.

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
