# Peer review of "Application of multimodality perception scene construction based on Internet of Things (IoT) technology in art teaching"

_PeerJ Computer Science, doi:10.7717/peerj-cs.2047_

## Round 0.1 · original submission · Major Revisions

Dear authors,

Thank you for submitting your article. The reviewers’ comments are now available. Your article has not been recommended for publication in its current form. However, we encourage you to address the reviewers' concerns and criticisms; particularly regarding readability, methods for experimental design and validity, and resubmit your article once you have updated it accordingly.

Best wishes,

**Language Note:** PeerJ staff have identified that the English language needs to be improved. When you prepare your next revision, please either (i) have a colleague who is proficient in English and familiar with the subject matter review your manuscript, or (ii) contact a professional editing service to review your manuscript. PeerJ can provide language editing services - you can contact us at [email protected] for pricing (be sure to provide your manuscript number and title). – PeerJ Staff

Reviewer 1 ·

Basic reporting

This study, exemplified by digital art flower cultivation, pioneers a multi-sensory channel interactive task integrating advanced technologies such as DenseNet networks for visual feature extraction and SoundNet convolutional neural networks for speech feature extraction. This innovative pedagogical model redefines art teaching, accentuating visual information while actively involving other senses as supportive participants. However, several areas requires improvement

Introduction Completeness: While the introduction is comprehensive, a summary of the author's contributions is lacking in the final section. Incorporating a concise overview of the author's unique contributions will enhance the introductory section's completeness.


Updated References: Several cited references are relatively old and may lack novelty. To bolster the article's currency, consider replacing some of the older references with more recent articles from reputable and innovative journals. This will enhance the relevance and reference value of the paper.

Experimental design

Formula Explanation: The introduction of formulas (4) through (7) in section 3.2 is currently insufficient. To enhance clarity, provide a more detailed explanation of the contents of these formulas. Consider adding context and elucidating their significance in the study.


Model System Experimental Parameters: Consider incorporating the experimental parameters of the model system at relevant points in the paper. Clearly specifying these parameters will contribute to the transparency and replicability of the study.

Validity of the findings

Usability Evaluation Experiment: In Section 4.2, the usability evaluation experiment and the extensive description of relevant time could be separated for a more organized presentation. This segregation would enhance clarity and facilitate a more focused understanding.

Conclusion Section Enhancement: The summary section appears somewhat redundant and lacks insights into the future development of the model methods. Enhance this section by providing a concise recapitulation and introducing potential avenues for further research and model refinement.

·

Basic reporting

This study endeavors to remedy deficiencies in art education through the implementation of a multi-modal perception system rooted in IoT technology. Utilizing interactive channels such as visual, auditory, tactile, and olfactory sensors, the system collects diverse data, including images, sounds, position, motion, and ambient light, thereby significantly elevating the quality of art teaching and optimizing learning efficiency. However, several aspects require further attention:
1. To enhance the completeness and richness of the abstract, consider rewriting it with a focus on expanding the sections related to research background, research content, and research results. This will provide a more comprehensive overview for readers.
2. The introduction should primarily highlight the background and research significance of the proposed model. Minimize excessive citations for comparative reference and concentrate on elucidating the unique contribution of the study in addressing the challenges in art education through the multi-modal perception system.

Experimental design

3. To enhance logical flow and rigor, introduce the application of art teaching within the framework of IoT technology and multi-modal perception scenarios. This will provide a clearer context for readers to understand the recent advancements in the field.
4. In Section 3.2, where the SoundNet network preprocesses an audio file three times, elucidate the distinct role of each preprocessing step. Clearly defining the purpose of each step will enhance transparency and understanding.

Validity of the findings

5. Provide a detailed introduction to the dataset and the data preprocessing process in the experimental analysis and discussion. This additional information is essential for readers to evaluate the robustness and reliability of the experimental results.
6. Specify the advantages of the proposed LSTM+SVM model in specific applications. Highlight how this model outperforms others in the context of art education, providing a clear rationale for its selection.

Additional comments

7. In the analysis and summary of the experimental section, ensure relevant references are included when comparing different model methods. This strengthens the credibility of the study and provides readers with a basis for understanding the comparative performance.
8. Strengthen the language professionalism throughout the manuscript to meet the standards required for publication. Ensure clarity, precision, and adherence to professional language conventions to enhance the overall quality of the paper.

---

## Round 0.2 · accepted · Accept

Dear authors,

Thank you for clearly addressing all the reviewers' comments. I confirm that the quality of your paper is improved. The paper is now ready for publication in light of this revision.

Best wishes,

Reviewer 1 ·

Basic reporting

Article structure, language, and sections are improved. Contribution is described clearly. Conclusion is presented in better way.

Experimental design

Methodology is improved with sufficient details. Formulas are presented and explained. Experimental parameters are now described.

Validity of the findings

Experimental results are described and presented in better way. Conclusion section is improved.

·

Basic reporting

After reading and revising of article, I show that all various required basic reporting comments are taking into account.

Experimental design

None

Validity of the findings

Variuous revisions are carried out.

Additional comments

None